# A Rapid Review of the Factors That Influence Service User Involvement in Interprofessional Education, Practice, and Research

**DOI:** 10.3390/ijerph192416826

**Published:** 2022-12-15

**Authors:** Michael Palapal Sy, Arden Panotes, Daniella Cho, Roi Charles Pineda, Priya Martin

**Affiliations:** 1National Teacher Training Center for the Health Professions, University of the Philippines Manila, Manila 1000, Philippines; 2Faculty of Medicine, Rural Clinical School, The University of Queensland, Toowoomba, QLD 4350, Australia; 3Department of Rehabilitation Sciences, Katholieke Universiteit Leuven, 3000 Leuven, Belgium; 4Health and Behavioral Sciences, The University of Queensland, Brisbane, QLD 4067, Australia

**Keywords:** consumer engagement, patient involvement, collaboration, interprofessional education and practice

## Abstract

Service user involvement in interprofessional education and collaborative practice remains limited despite the increasing push for this by governments and grant funding bodies. This rapid review investigated service user involvement in interprofessional education, practice, and research to determine factors that enable or hinder such involvement. Following the Cochrane and the World Health Organization’s rapid review guidelines, a targeted search was undertaken in four databases. Subsequent to the screening processes, included papers were critically appraised, and extracted data were synthesized narratively. Sixteen studies met inclusion criteria. Most studies were related to interprofessional collaborative practice, as opposed to education and research. Service user involvement was more in the form of consultation and collaboration, as opposed to consumer-led partnerships. Enablers and barriers to service user involvement in IPECP were identified. Enablers included structure, the valuing of different perspectives, and relationships. Barriers included time and resources, undesirable characteristics, and relationships. This rapid review has added evidence to a swiftly expanding field, providing timely guidance. Healthcare workers can benefit from targeted training. Policy makers, healthcare organizations, and governments can investigate strategies to mitigate the time and resource challenges that impede service user involvement in IPECP.

## 1. Introduction

Service user involvement in healthcare education, practice, and research has gained attention in response to government policies that mandate client and public involvement in healthcare services [1,2]. Service users are those who use or are affected by the services provided by healthcare workers, including clients and their carers [3]. Often, service user involvement in healthcare is also referred to as consumer involvement [4], patient and consumer engagement [5], patient involvement [6], and patient participation [7]. In principle, it embodies person-centered care where service users are included in decision making related to the services they receive. Services that have engaged service users in decision making have achieved higher levels of patient satisfaction and improved healthcare outcomes [8].

Service user involvement can also be useful in education and research. The recognition that service users are experts of their own experience [9] places value on their contribution to the education of healthcare workers. Service user involvement in health professions education literature has included membership in advisory groups, sharing personal stories during classroom discussions, and assessing student performance during fieldwork placements [10]. This involvement in education has been documented to be of remarkable benefit to service users, as taking part in teaching activities can empower their sense of value [11]. A rapid review by Slattery et al. [12] revealed that service user involvement in research includes a wide range of activities such as occasional end-user feedback on research materials, power-sharing arrangements in research activities, and end-user-led research. These activities can be paralleled by the three steps of participation proposed by the Consumer in NHS Research: consultation (e.g., asking service users for their views that will inform decision making), collaboration (e.g., active, on-going partnership with consumers such as being a committee member in a task force), and consumer-controlled research (e.g., consumers designing, undertaking, and disseminating research results; healthcare workers being involved only upon invitation from the service users) [13].

Interprofessional education (IPE) programs, where students learn with, from, and about each other [14], can improve students’ understanding of their own role [15], enhance positive attitudes towards other professionals [16], and teamwork skills [17]. These skills are necessary competencies for working in interprofessional teams to deliver person-centered care. Students that have educational experiences with service users have been shown to develop a better understanding of the patient-centered perspective and gain skills to work effectively in an interprofessional environment [18]. Interprofessional collaborative practice initiatives can be valuable to practicing (i.e., post-qualification/registration) healthcare workers [19]. Therefore, Interprofessional Education and Collaborative Practice (IPECP) initiatives, given their person-centered nature, are well-placed for service user involvement right from the service or project initiation phase.

In practice, service user involvement in IPECP is limited due to several challenges. A review by Repper and Breeze [10] showed that there are few IPE initiatives that involve service users. Educational experiences that involve service users are mostly in the community mental health field and are small-scale qualitative studies that evaluate the impact of the experience on students and service users [10,18,19]. An interview study by Kvarnström and colleagues [20] showed that healthcare workers have a varied understanding of service user involvement in an interprofessional context. They identified that service user involvement can cause a possible power imbalance between the healthcare worker and the service user. IPECP research in the real world can be hard to implement and sustain, given the multiple layers involved [21]. This may make it less appealing to involve another stakeholder group (i.e., services users) in the design and conduct of research. IPECP experts advocate for the use of innovative approaches that include service users in research as a priority agenda [21]. The research trend in IPECP is moving towards enhancing the relevance of the research to those who are impacted by the outcomes, with attention given to diversity and inclusivity [22]. Globally, funding bodies are increasingly pushing for service users to be involved in research projects [23,24]. However, within the IPECP field, little is known about the extent and nature of service users’ involvement in education, practice, and research initiatives.

### 1.1. Review Aims

The aim of this rapid review is to investigate service user involvement in interprofessional education, practice, and research, and to determine factors that enable or hinder.

### 1.2. Review Questions

What is the current status of service user involvement in interprofessional education, practice, and research?What are the enablers of and barriers to service user involvement in interprofessional education, practice, and research?

## 2. Materials and Methods

In line with the resources available and to swiftly add evidence in this rapidly expanding field, a rapid review method was chosen. The study followed the guidelines for conducting rapid reviews proposed by the World Health Organization [25] and Cochrane [26]. The steps of the review included needs assessment, protocol development, the literature search, screening and study selection, data extraction, risk of bias assessment, knowledge synthesis and report dissemination [25]. Expert consultation with IPECP stakeholders was also conducted while developing the protocol. The WHO checklist for rapid reviews was used to ensure quality assurance (Appendix A).

### 2.1. Needs Assessment

To initiate the needs assessment, a preliminary literature search was conducted by the reviewers to determine the suitability of the topic’s scope and the appropriate methods. The review team undertook this review in response to one of the research priorities of IPECP, which is to promote the participation and involvement of service users in IPECP-related education, practice, and research, to influence policy making [27].

### 2.2. Protocol Development

A study protocol was developed to guide the review and was registered on Open Science Framework (https://osf.io/p5f2w/) (accessed on 8 November 2022).

### 2.3. Literature Search

#### 2.3.1. Database Selection

To yield a manageable number of studies for this review considering the available resources, the electronic databases searched were limited to Cochrane Central, MEDLINE via Ovid, Embase, and CINAHL via EBSCO. These databases were chosen as they include multidisciplinary journals on health and social care, and education.

#### 2.3.2. Search Strategy

The population, intervention, comparator, outcome (PICO) elements (Table 1) were used to formulate the key terms for the search strategy and develop the inclusion criteria for study selection. Based on PICO, the key terms for the search were service users (population), interprofessional education/practice/research (investigated phenomenon) and involvement (outcome). The search string was optimized for each database and developed from a combination of controlled vocabularies (e.g., MeSH and Emtree) and free text search (limited to title, abstract and keywords search) that included the most pertinent related terms. A search limit was applied to identify only English-language peer-reviewed articles published from 2006 onwards. Furthermore, only articles with an abstract were searched to exclude other types of publication like commentaries and editorial letters. A specialist subject librarian was consulted during the development of the search string. The full search strings used for each database have been included in Appendix A.

### 2.4. Screening and Study Selection

Records yielded by the previous step were transferred to Endnote X9 (Clarivate, Philadelphia, PA, USA) and uploaded onto Covidence for deduplication and screening. Cochrane rapid review guidelines [26] were followed in a two-stage screening process. During the title and abstract screening stage, all the reviewers screened 20 records together to pilot the application of the inclusion/exclusion criteria for screening. Thereafter, two reviewers (AP and DC) dual screened 20% of the records (*n* = 645), with conflict resolution by the rest of the reviewers, to establish consistency. Finally, the remaining records were split between the two reviewers (AP and DC). In the full-text screening stage, all reviewers screened ten articles together to standardize the process. The full-text screening of the remaining articles was completed by two reviewers (AP and DC).

### 2.5. Data Extraction

A customized data extraction form was developed, piloted, and refined by the entire review team. Two reviewers (AP and DC) extracted pertinent data from the included articles. As a group, all reviewers verified the tabulated data for correctness and completeness. See Appendix A for the data extraction template used.

### 2.6. Risk of Bias (Quality) Assessment

Critical appraisal of the included papers was completed by three reviewers (AP, DC, and RCP), with conflict resolution provided by a fourth reviewer (MPS). The fourth reviewer (MPS) also cross-checked the results of the review using six randomly selected papers for quality assurance. The modified McMaster critical appraisal tool for quantitative [28] and qualitative [29] studies and Hong et al.’s [30] Mixed Methods Appraisal Tool were used. These tools were chosen (over the JBI tools as outlined in the protocol) as they were determined by the review team as more suited to the types of studies included in the review. In addition, these tools are freely available and widely used in similar reviews.

### 2.7. Knowledge Synthesis

A narrative synthesis of the identified literature was conducted to develop themes [31]. The papers were grouped based on the variables used for data extraction. The narrative synthesis reported the results of included studies and discussed the reasons for differences among studies, such as heterogeneity of the PICO elements, study design, and methodological quality. The final report includes implications of the review findings, and recommendations for policy and practice.

## 3. Results

The database search yielded 4359 records. After removal of 1128 duplicates, 3231 records were subjected to title and abstract screening. Subsequently, 148 articles were retrieved for full-text review. A total of 114 records were excluded because of wrong intervention. Nine records were excluded because of wrong population. Seven records were excluded because of wrong study design and two were excluded because full text articles were not available. Finally, 16 articles met all the inclusion criteria and were included in this review. Further information is presented in Figure 1 (a flow diagram of included studies). Canada [32,33,34] and the Netherlands [35,36,37] produced three publications each. Germany [38,39] and the UK [40,41] had two publications each. Australia [42], Norway [43], Sweden [44], Switzerland [45], the USA [46], and Canada and the USA combined [47] had one publication each.

Included studies were published between 2006 to 2021. The years 2006 [41], 2012 [40], and 2013 [38] had one publication each. A spike to two to four publications was observed in 2014 [39,44], 2015 [34,42,43,47], and 2017 [35,36]. Subsequently, there was one publication in 2018 [37], and two in 2019 [33,45] and 2021 [32,46]. Overall, the number of publications on this topic over the last ten years appears to have risen and peaked in 2015, after which the numbers have lowered and remained steady.

A majority (*n* = 10) of the publications reported IPECP practice initiatives, followed by IPECP practice and research initiatives (*n* = 3), IPECP practice and education initiatives (*n* = 2), and an IPECP education initiative (*n* = 1). Study settings included primary care services, rehabilitation clinics, social services, chronic disease management services, ICUs, obstetrics networks, child welfare services, cancer networks and home hospices. Patients in the included studies were children and young people, pregnant women, those with chronic disease conditions, those in ICUs or those serviced by cancer networks, and community care and family health. Service users in the included studies were partners, family members in a caring role, and community care providers and volunteers. Healthcare workers in the included studies consisted of general practitioners, family physicians, IT professionals, nurses, patient navigators, patient advocates, pharmacists, physiotherapists, practice managers, social workers, speech pathologists, and researchers.

The participant sample size reported in the included studies ranged from five in an interview study [43] to over 629 participants in a randomized controlled trial (RCT) [39]. The age range of participants in the included studies was from five to ten years [44] to 87 years [45]. Most studies used qualitative methods (*n* = 14), followed by mixed methods (*n* = 1) and quantitative methods (*n* = 1), specifically a multi-center cluster RCT [38]. The qualitative methods used were largely interviews (*n* = 7), focus groups (*n* = 5), observation (*n* = 3), ethnography (*n* = 1), document review (*n* = 1), participatory action research (*n* = 1), and the persona-scenario method (*n* = 1). Mixed methods consisted of focus groups and expert surveys. Further information about the included studies has been presented in Table 2 (Study characteristics).

### 3.1. Methodological Quality

The included studies all clearly stated the study purpose and provided a relevant review of the literature. One study did not thoroughly describe the process of study selection [34]. Seven studies utilized an inductive approach to data analysis [32,33,37,41,43,45,46]. Only one qualitative study identified the assumptions and biases of the researcher [32]. All studies presented findings that were consistent and reflective of the gathered data. However, there are two studies which did not adequately describe the process of analyzing data [34,40]. The quantitative study and all qualitative studies provided appropriate conclusions given the study methods and results. The modified McMaster critical appraisal checklist for qualitative and quantitative studies and the Mixed Methods Appraisal Tool are attached as Appendix A.

### 3.2. Service User Involvement in IPECP Education, Practice, and Research

#### 3.2.1. Nature of Service User Involvement

Studies reported service user involvement in IPECP practice, training, and research. Involvement of service users in interprofessional meetings was reported frequently, such as intake and discharge planning meetings [35,36,44,45], as well as models of care where service users were considered as part of the interprofessional team [32,33]. Studies also discussed co-creation of a shared decision-making model [37] and shared decision-making training programs [38,39]. In the study by Reeves et al. [47], service users were involved in ward rounds to enable shared decision making. Other studies involved service users in workshops and training [34], and in quality improvement projects [34,40]. The study by Sæbjørnsen and Willumsen [43] engaged service users in a variety of ways including case conferences, evaluations, and roundtable discussions. Some studies outlined the involvement of service users in novel strategies such as a partnership group or model of care related to cancer care [41], and engagement in medication review [46].

#### 3.2.2. Enablers

Enablers of service user involvement in IPECP identified in the included studies can be summarized into three categories: structure, valuing different perspectives, and relationships.

*Structure*: Having a structure to the involvement of service users in IPECP initiatives can promote intentionality [44,45] and enable healthcare workers to make time to engage with this [34]. Use of a specific framework can also assist with providing a structure [47]. Funding can also enable healthcare workers to invest time in service user involvement [42]. One study found that use of video vignettes enhanced participant focus in interprofessional team meetings [35].

*Valuing different perspectives*: There needs to be a desire for and willingness to engage service users in IPECP initiatives [32,40,42]. This was facilitated by active listening, active participation [34] and patient involvement in evaluations [32,38]. Co-creating with service users [37] and active participation of patients [36] were considered important.

*Relationships*: Participants in the included studies reiterated the importance of building trust [43] through positive interactions [45], which can lead to collaborative partnerships [41,43].

#### 3.2.3. Barriers

Barriers identified in the included studies can be summarized into three categories: time and resources, undesirable characteristics, and relationships.

*Time and resources*: Service user involvement in IPECP can be limited by a lack of time on the part of healthcare workers [32,33,34,38,39] and patients [32,40]. Additionally, a lack of resources including costs to cover parking, transport, and food, as well as institutional commitment [33,41] can hamper this.

*Undesirable characteristics*: Negative attitudes such as lack of inclusivity, inflexibility, offensive remarks, use of jargon, and lack of knowledge and skills on the part of patients [36] and healthcare workers [32,37,38,44] can impede collaboration and thus service user involvement in IPECP. Sub-cultures and conflicts within teams and between professions can also be a significant barrier [35,37,38,39,47].

*Relationships*: Poor relationships between patients and their care providers [42], conflicts [45], distrust in healthcare workers due to breach in confidentiality [43], and service users being a vulnerable group (e.g., palliative care patients; [46]) can all impede engagement and participation in IPECP.

## 4. Discussion

This rapid review investigated the evidence on service user involvement in IPECP practice, education, and research. Service user involvement in included studies was most predominant in practice and less common in teaching, education, and research. Most included studies were conducted *on* service users, thereby only enabling consultation and collaboration levels of participation, as opposed to consumer-controlled or consumer-led initiatives [13]. Cancer Australia’s [48] levels of consumer involvement framework provides healthcare workers and researchers with guidance on maximizing the level of service user engagement and participation in health care, with applicability to research and education as well. This review highlights the need to include service users as partners right from project conceptualization to completion. Being more cognizant of existing frameworks on levels of service user engagement will enable researchers and program developers to become more intentional in co-designing IPECP initiatives.

Most included studies in this review originated from Western countries. This could be because Western principles of individualism, self-actualization, and the liberty to exercise human rights underpin patient autonomy and agency [49]. Service user involvement in practice, education, and research reflects principles of inclusivity, respect, participation, iteration, and being outcome-focused [50]. These principles are only emerging in populations that embrace collectivistic cultures where decision making and participation are dependent on interpersonal, intergenerational, and familial considerations [51], such as those seen in low- and middle-income countries. This review highlights the need for promoting service user involvement internationally, as it embodies good use of research and prevents wastage of public research funding [52,53]. Good use of research will in turn enhance healthcare outcomes of peoples, communities, and populations.

Participatory action research and qualitative methods can facilitate service user engagement in research, as these approaches can bring the lived experiences of service users to the foreground [54,55]. The nature of relationships between a healthcare worker and consumer can either enable or hinder consumer engagement. Therefore, both parties need to strive to build a positive and supportive partnership. Moreover, by staying true to the epistemological and ontological underpinnings of participatory action research, the co-design process with service users can intentionally expand from clinical practice to health professions education, training, and research. Following Towle’s [9] taxonomy on the degree of involvement of service users in teaching and learning encounters, educators can co-design curriculums and instructions not just for but with service users. Professional development courses on maximizing service user involvement in IPECP, including the use of recommended research approaches, are warranted to upskill healthcare workers to further engage with service users.

It is noteworthy to highlight the challenges faced by service users with cognitive impairments or high-risk health conditions [46] noted in this review. Acknowledging the difficulties in engaging specific service user populations in the co-design process, several researchers have provided frameworks and recommendations to mitigate these challenges [56,57,58,59]. Although not all service users are willing or able to be involved in their own care, it is essential that service providers (i.e., healthcare workers) do not act as gatekeepers of the process. Rather than making assumptions or withholding options, service providers should support individual autonomy and leave the decision to participate to the service users themselves [57]. For those who are most cognitively or physically impaired, service providers should recognize the value of insights of families, friends, and advocates who have first-hand knowledge of the issues that matter to people with severe impairments [56]. There is also a need to advance this body of work in pediatric and adolescent services so that consumers in these areas can also be active participants in their care. Policy makers and healthcare organizations need to invest in this area to make it easier for service users with significant impairments to engage with service providers to enhance their healthcare journey.

### Strengths and Limitations

This review has added evidence to a field that is increasingly getting more attention from governments and grant funding bodies. It provides evidence to support the need for further work in the area of service user involvement in IPECP. Despite being a rapid review, this research included results from four database searches and was inclusive of a broad range of research studies utilizing quantitative, qualitative, and mixed methods designs. Using a comprehensive range of critical appraisal tools enabled appraisal of all the studies appropriate to the study designs used. Reviewers in this team have collective expertise in IPECP (MPS, RCP, PM) and review methodology (MPS, RCP, PM). They possess experience in diverse health systems across Australia (DC, PM), Belgium (RCP), India (PM), Japan (MPS) and the Philippines (MPS, AP), in a variety of roles including clinical (all reviewers), teaching (MPS, PM) and research roles (all reviewers), and represent two professions: occupational therapy (MPS, AP, RCP, PM) and medicine (DC). These experiences guided the iterative process of fine-tuning the PICO, as well as interpreting the results of the review from different perspectives and contexts. Use of the Cochrane and WHO rapid review guidelines ensured the rigor of the review processes. In line with the resources available, the review team followed strict criteria to only include studies that explicitly used the term ‘interprofessional’. This may have excluded other IPECP studies that used different terminology to denote the same concept, which is reflective of the wider terminology issue in this field. Further systematic reviews are needed in this area to provide a more comprehensive view of this topic. This review team did not include a consumer. Future reviews can include a consumer on the review team to enable interpretation of findings also through a consumer lens. 

## 5. Conclusions

This rapid review has added evidence to a rapidly expanding field, providing timely guidance on service user involvement in IPECP practice, education, and research. The available evidence indicates that service users are most engaged in IPECP practice, in comparison to education and research. Furthermore, the levels of engagement are mainly at the lower levels of the engagement hierarchy, as well as more towards the tail end of a project. This review highlights the need for healthcare workers to engage service users right from project initiation in practice, education, and research, plus work towards higher levels of engagement moving towards consumer-led partnerships. Healthcare workers can benefit from targeted training that provides them with a skillset in co-designing IPECP initiatives with service users right from the outset. Policy makers, healthcare organizations, and governments can investigate strategies to mitigate the time and resource challenges that impede service user involvement in IPECP practice, education, and research. Further research can investigate ways to promote higher levels of engagement of service users in education and research, including service users with significant cognitive, physical, or mental impairments.

## Figures and Tables

**Figure 1 ijerph-19-16826-f001:**
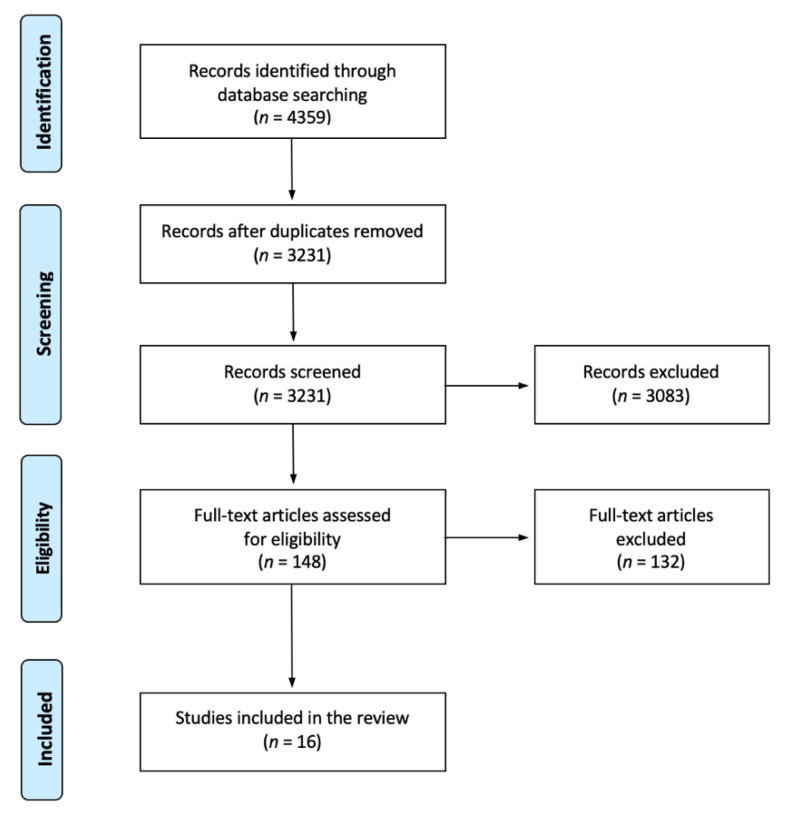
Flow diagram of included studies.

**Table 1 ijerph-19-16826-t001:** Inclusion and exclusion criteria.

	Inclusion Criteria	Exclusion Criteria
Population	Patients or service users who have received a healthcare service from a healthcare worker and/or their carer/s who have participated in an IPECP initiative with a healthcare worker	Patients or service users who were not involved in an IPECP initiative with a healthcare worker
Intervention or investigated phenomena	IPECP initiatives for entry-level and practicing professionals that involve healthcare workers from at least two professions working together with service user/s	Education, research, and practice activities that are labelled as multidisciplinary, interdisciplinary, or transdisciplinary
Comparator	No comparator	No comparator
Outcome	Description of the involvement of service users in the IPECP initiatives	Outcomes that are not attributed to the involvement of service users
Research designs	Primary research: quantitative, qualitative, and mixed methods designs	Secondary research (i.e., reviews), conference abstracts/posters, study protocols, editorials/commentaries, position papers
Other	English-language literature published from 2006 onwards	Unpublished literature (e.g., grey literature, theses, and dissertation manuscripts)

**Table 2 ijerph-19-16826-t002:** Study characteristics.

No.	Author/s and Year	Design	Country	Participants		Measures	Setting
				Professionals	Service Users		
1	Bolin, 2014 [44]	Qualitative	Sweden	Social workers and professionals in child psychiatry	Children receiving social services (*n* = 28)	Attendance in collaborative meetings	Practice
2	Carr et al., 2012 [40]	Qualitative	United Kingdom	General practitioners, nurses, physiotherapists, and managers (*n* = 44)	Clients with back pain (*n* = 11)	Participation in workshops for quality	Education
3	Koerner et al., 2014 [39]	Quantitative	Germany	Physicians, nursing staff, physical therapists, sport teachers, masseurs, psychologists and other psychosocial therapists, dietitians, and social workers	Clients with chronic disease	Participation on a survey for a training program evaluation	Practice
4	Körner et al., 2013 [38]	Mixed methods	Germany	Physicians, nursing staff, physical therapists, sport teachers, masseurs, psychologists and other psychosocial therapists, dietitians, and social workers (*n* = 32)	Rehabilitation clients (*n* = 36)	Involvement in focus group discussions in developing a training program for health professionals	Practice
5	Metersky et al., 2021 [32]	Qualitative	Canada	Nurses, social workers, dietitians, pharmacists, a nurse practitioner, and a respiratory therapist (*n* = 10)	Clients with chronic disease diagnosis (*n* = 10)	Participation in group interviews for designing interprofessional teams	Practice/research
6	Molenaar et al., 2018 [37]	Qualitative	Netherlands	Primary care midwives, hospital-based midwives, obstetricians, obstetric nurses, and maternity care assistants	Pregnant women and their partners (*n* = 71)	Involvement in co-creating a shared decision-making model	Practice
7	Phillips et al., 2015 [42]	Qualitative	Australia	Nurses, physiotherapists, exercise physiologists, fitness instructors, social workers, and general practitioners (*n* = 14)	Clients with chronic disease and their carers (*n* = 55)	Participation in interviews to describe ‘patient as professional’ role	Practice
8	Reeves et al., 2015 [47]	Qualitative	USA and Canada	Nurses, doctors, pharmacists, and social workers	Clients from the intensive care unit	Involvement of family members in co-designing the treatment plan	Practice/education
9	Sæbjørnsen & Willumsen, 2015 [43]	Qualitative	Norway	Social workers, childcare specialists, and therapists	Children receiving social welfare services	Participation in children conferences	Practice
10	Schoeb et al., 2019 [45]	Qualitative	Switzerland	Doctors, nurses, physiotherapists, occupational therapists, social workers, psychologists, speech therapists, dieticians, and work rehabilitation staff	Clients with various conditions (*n* = 25)	Participation in interprofessional meetings for discharge planning	Practice
11	Sitzia et al., 2006 [41]	Qualitative	United Kingdom	Nurses, doctors, administrators, and managers	Clients with cancer (*n* = 59)	Involvement in project evaluation through interviews	Practice
12	Tjia et al., 2021 [46]	Qualitative	USA	Hospice administrator, nurses, physicians, pharmacists (*n* = 8)	Former family caregivers (*n* = 10)	Involvement in stakeholder panel meetings	Practice
13	Valaitis et al., 2019 [33]	Qualitative	Canada	Healthcare providers/community care providers (*n* = 29), community service providers (*n* = 12), volunteers (*n* = 14)	Clients with complex and chronic conditions (*n* = 70)	Involvement in designing a primary care service	Practice
14	van Dongen et al., 2017 [36]	Qualitative	Netherlands	Healthcare professionals (*n* = 8)	Clients with chronic conditions and their relatives (*n* = 11)	Participation in interprofessional meetings	Practice
15	Van Dongen et al., 2017 [35]	Qualitative	Netherlands	Family medicine and occupational therapist	Clients with chronic conditions (*n* = 7)	Participation in interprofessional team meetings	Practice/research
16	Worswick et al., 2015 [34]	Qualitative	Canada	Primary healthcare professionals	Clients with back pain (*n* = 11)	Participation in workshops	Practice/education

## Data Availability

The review did not report any data.

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
