# Peer review of "A Rapid Review of the Factors That Influence Service User Involvement in Interprofessional Education, Practice, and Research"

_ijerph, 2022, doi:10.3390/ijerph192416826_

Round 1

Reviewer 1 Report

Thank you for allowing me to review your very interesting rapid review onf the factors that influence service user involvement in interprofessional education, practice, and research.

I enjoyed reading the review very much. Your describtion of screening and selection of studie is clear and concise.

I would appreciate if you could elaborate on some minor points.

1) The choice of method. Why did you apply the rapid review methodology as opposed to for example the scoping review?

2) Risk of bias assessment

Besides being freely available and otherwise used, what did the tools that you used provide for your assessment? Perhaps this could be covered in the strength and limitations section.

3) Regarding 2.7 Knowledge synthesis

The title is knowledge synthesis, however you use the word narrative synthesis in the text. Please elaborate on how this synthesis was conducted and how did it as synthesis rarely conduct themselves.

4) Records excluded: With such a large volumen of excluded studies could you categorize them?

5) relationships are present in both enablers and barriers - could you reflect on that in the discussion?

6) The discussion is very interesting and brings to the table many good points on culture, and participatory research methods. Besides highlighting the challenges of service users with cognitive impairments and high-risk conditions, there is the large group of paediatric patients and adolescents who are also a particular segment of service users.

Reviewer 2 Report

Thanks for the opportunity to review you submission. In the main this is a well structured and written review. I hope my comments help to enhance your work.

You describe this as a rapid review. Firstly you then describe a narrative review-even methods for analysis-I recommend this is in the abstract as well. Do you need to define the different review methods and how they apply.

Your limitations and strengths is all about strengths. Surely a rapid review does not have the same credibility as a full systematic review-please discuss this.

Bias in the reviewers you were confident that your roles negated this. Would it have been appropriate to have a service user co participant or adviser. Was your lens from a practitioner/educator? 
